# A Linearly Convergent Proximal Gradient Algorithm for Decentralized Optimization

**Sulaiman A. Alghunaim, Kun Yuan**
Electrical and Computer Engineering Department
University of California Los Angeles
Los Angeles, CA, 90095
{salghunaim,kunyuan}@ucla.edu

**Ali H. Sayed**
Ecole Polytechnique Fédérale de Lausanne
CH-1015 Lausanne, Switzerland
ali.sayed@epfl.ch

## Abstract

Decentralized optimization is a powerful paradigm that finds applications in engineering and learning design. This work studies decentralized composite optimization problems with non-smooth regularization terms. Most existing gradient-based proximal decentralized methods are known to converge to the optimal solution with sublinear rates, and it remains unclear whether this family of methods can achieve global linear convergence. To tackle this problem, this work assumes the non-smooth regularization term is common across all networked agents, which is the case for many machine learning problems. Under this condition, we design a proximal gradient decentralized algorithm whose fixed point coincides with the desired minimizer. We then provide a concise proof that establishes its linear convergence. In the absence of the non-smooth term, our analysis technique covers the well known EXTRA algorithm and provides useful bounds on the convergence rate and step-size.

## 1 Introduction

Many machine learning problems can be cast as composite optimization problems of the form

$$\min_{w\in\mathbb{R}^M}\; J(w) + R(w), \quad \text{where} \quad J(w) = \frac{1}{N}\sum_{n=1}^{N}Q(w;x_n) \tag{1}$$

where $w$ is the optimization variable, $x_n$ is the $n$-th data, and $N$ is the size of the dataset. The loss function $Q(w;x_n)$ is assumed to be smooth, and $R(w)$ is a regularization term possibly non-smooth. Typical examples of $R(w)$ can be the $\ell_1$-norm, the elastic-net norm, and indicator functions of convex sets (e.g., non-negative orthants). Problems of the form (1) arise in different settings including, among others, in model fitting [1] and economic dispatch problems in power systems [2].

When the data size $N$ is very large, it becomes intractable or inefficient to solve problem (1) with a single machine. To relieve this difficulty, one solution is to divide the $N$ data samples across multiple machines and solve problem (1) in a cooperative manner. Many useful distributed algorithms exist that solve problem (1) across multiple computing agents such as the distributed alternating direction method of multipliers (ADMM) [1, 3], parallel SGD methods [4, 5], distributed second-order methods [6–8], and parallel dual coordinate methods [9, 10]. All these methods are designed for a centralized network topology, e.g., parameter servers [11], where there is a central node connected to all computing agents that is responsible for aggregating local variables and updating model parameters. The potential bottleneck of the centralized network is the communication traffic jam on the central node [12–14]. The performance of these non-decentralized methods can be significantly degraded when the bandwidth around the central node is low.

In contrast, decentralized optimization methods are designed for any connected network topology such as line, ring, grid, and random graphs. There exists no central node for this family of methods, and each computing agent will exchange information with their immediate neighbors rather than a remote central server. Decentralized methods to solve problem (1) have been widely studied for some time in the signal processing, control, and optimization communities [14–28]. More recently, there have been works in the machine learning community with interest in these problems due to their advantages over centralized methods [12, 13, 29, 30]. Specifically, since the communication can be evenly distributed across each link between nodes, the decentralized algorithms converge faster than centralized ones when the network has limited bandwidth or high latency [12, 13].

For the smooth case, the convergence rates of decentralized methods are comparable to centralized methods. For example, the decentralized methods in [15, 27, 31] are shown to converge at the sublinear rate $O(1/i)$ (where $i$ is the iteration index) for smooth and convex objective functions, and at the linear rate $O(\gamma^i)$ (where $\gamma \in (0, 1)$) for smooth and strongly-convex objective functions. These convergence rates match the convergence rates of centralized gradient descent. However, some gap between decentralized and centralized *proximal* gradient methods continues to exist in the presence of a composite non-smooth term. While centralized proximal gradient methods have been shown to converge linearly when the objective is strongly convex [32], it remains an open question to establish the linear convergence of *decentralized* proximal gradient methods. This work closes this gap by proposing a proximal gradient decentralized algorithm that is shown to converge linearly to the desired solution. Next we explain the problem set-up and comment on existing related works.

## 1.1 Problem Set-up

Consider a network of $K$ agents (e.g., machines, processors) connected over some graph. Through only local interactions (i.e., agents only communicate with their immediate neighbors), each node is interested in finding a consensual vector, denoted by $w^\star$, that minimizes the following aggregate cost:

$$w^\star = \underset{w \in \mathbb{R}^M}{\arg\min} \quad \frac{1}{K} \sum_{k=1}^{K} J_k(w) + R(w) \tag{2}$$

The cost function $J_k(w) : \mathbb{R}^M \to \mathbb{R}$ is privately known by agent $k$ and $R(w) : \mathbb{R}^M \to \mathbb{R} \cup \{\infty\}$ is a proper[1] and lower-semicontinuous convex function (not necessarily differentiable). When $J_k(w) = \frac{1}{L} \sum_{n=1}^{L} Q(w; x_{k,n})$ where $\{x_{k,n}\}_{n=1}^{L}$ is the local data assigned or collected by agent $k$, and $L$ is the size of the local data, it is easy to see that problem (2) is equivalent to its centralized counterpart (1) for $N = KL$. We adopt the following assumption throughout this work.

**Assumption 1. (Cost function)**: There exists a solution $w^\star$ to problem (2). Moreover, each cost function $J_k(w)$ is convex and first-order differentiable with $\delta$-Lipschitz continuous gradients:

$$\|\nabla J_k(w^o) - \nabla J_k(w^\bullet)\| \leq \delta \|w^o - w^\bullet\|, \quad \text{for any } w^o \text{ and } w^\bullet \tag{3}$$

and the aggregate cost function $\bar{J}(w) = \frac{1}{K} \sum_{k=1}^{K} J_k(w)$ is $\nu$-strongly-convex:

$$(w^o - w^\bullet)^\mathsf{T} \left( \nabla \bar{J}(w^o) - \nabla \bar{J}(w^\bullet) \right) \geq \nu \|w^o - w^\bullet\|^2 \tag{4}$$

for any $w^o$ and $w^\bullet$. The constants $\nu$ and $\delta$ satisfy $0 < \nu \leq \delta$. □

Note that from the strong-convexity condition (4), we know the objective function in (2) is also strongly convex and, thus, the global solution $w^\star$ is unique.

## 1.2 Related Works and Contribution

Research on decentralized/distributed optimization and computation dates back several decades (see, e.g., [33–36] and the references therein). In recent years, various centralized optimization methods such as (sub-)gradient descent, proximal gradient descent, (quasi-)Newton method, dual averaging, alternating direction method of multipliers (ADMM), and other primal-dual methods have been extended to the decentralized setting. The core problem in decentralized optimization is to design methods with convergence rates that are comparable to their centralized counterparts. For the smooth

case ($R(w) = 0$), the decentralized primal methods from [16, 37–39] converge linearly to a *biased* solution and not the exact solution. For convergence to the exact solution, these primal methods require employing a decaying step-size that slows down the convergence rate making it sublinear at $O(1/i)$ in general. The works [18–22] established linear convergence to the *exact* solution albeit for decentralized primal-dual methods based on ADMM or inexact augmented Lagrangian techniques. Other works established linear convergence for simpler implementations including EXTRA [15], gradient tracking methods [26, 27], exact diffusion [28], NIDS [40], and others. The work [30] study the problem from the dual domain and propose accelerated dual gradient descent to reach an optimal linear convergence rate for smooth strongly-convex problems.

There also exist many works on decentralized composite optimization problems with non-smooth regularization terms. The work [41] considered a similar set-up to this work and proposed a proximal gradient method combined with Nesterov's acceleration that can achieve $O(1/i^2)$ convergence rate; however, it requires an increasing number of inner loop consensus steps with each iteration leading to an expensive solution. Other works focused on the case where each agent $k$ has a local regularizer $R_k(w)$ possibly different from other agents. For example, a proximal decentralized linearized ADMM (DL-ADMM) approach is proposed in [22] to solve such composite problems with convergence guarantees, while the work [42] establishes a sublinear convergence rate $O(1/i)$ for DL-ADMM when each $J_k(w)$ is smooth with Lipschitz continuous gradient. PG-EXTRA [23] extends EXTRA [15] to handle non-smooth regularization local terms and it establishes an improved rate $o(1/i)$. The NIDS algorithm [40] also has an $o(1/i)$ rate and can use larger step-sizes compared to PG-EXTRA. Based on existing results, there is still a clear gap between decentralized algorithms and centralized algorithms for problem (2) when using proximal gradient methods.

The work [43] established the *asymptotic* linear convergence[2] of a proximal decentralized algorithm for the special case when *all* functions $\{J_k(w), R_k(w)\}$ (possibly different regularizers) are *piecewise linear-quadratic* (PLQ) functions. While this result is encouraging, it does not cover the *global* linear convergence rate we seek in this work since their linear rate occurs only after a sufficiently large number of iterations and requires all costs to be PLQ. Another useful work [29] extends the CoCoA algorithm [9] to the COLA algorithm for decentralized settings and shows linear convergence in the presence of a non-differentiable regularizer. Like most other dual coordinate methods, COLA considers decentralized learning for *generalized linear models* (e.g., linear regression, logistic regression, SVM, etc). This is because COLA requires solving (2) from the dual domain and the linear model facilitates the derivation of the dual functions. Additionally, different from this work, COLA is not a proximal gradient-based method; it requires solving an inner minimization problem to a satisfactory accuracy, which is often computationally expensive but necessary for the linear convergence analysis.

Note finally that decentralized optimization problems of the form (2) can be reformulated into a consensus equality constrained optimization problem (see equation (7)). The consensus constraint can then be added to the objective function using an indicator function. Several works have proposed general solutions based on this construction using proximal primal-dual methods – see [44–46] and references therein. Linear convergence for these methods have been established under certain conditions that do not cover decentralized composite optimization problems of the form (2). For example, the works [44, 45] require a smoothness assumption, which does not cover the indicator function needed for the consensus constraint. The work [46] requires the coefficient matrix for the non-smooth terms to be full-row rank, which is not the case for decentralized optimization problems even when $R(w) = 0$.

**Contribution.** This paper considers the composite optimization problem (2) and has two main contributions. First, for the case of a common non-smooth regularizer $R(w)$ across all computing agents, we propose a proximal decentralized algorithm whose fixed point coincides with the desired global solution $w^\star$. We then provide a short proof to establish its linear convergence when the aggregate of the smooth functions $\sum_{k=1}^{K} J_k(w)$ is strongly convex. This result closes the existing gap between decentralized proximal gradient methods and centralized proximal gradient methods. The second contribution is in our convergence proof technique. Specifically, we provide a concise proof that is applicable to general decentralized primal-dual gradient methods such as EXTRA [15] when $R(w) = 0$. Our proof provides useful bounds on the convergence rate and step-sizes.

**Notation.** For a matrix $A \in \mathbb{R}^{M \times N}$, $\sigma_{\max}(A)$ ($\sigma_{\min}(A)$) denotes the maximum (minimum) singular value of $A$, and $\underline{\sigma}(A)$ denotes the minimum *non-zero* singular value. For a vector $x \in \mathbb{R}^M$ and a positive semi-definite matrix $C \geq 0$, we let $\|x\|_C^2 = x^\mathsf{T} C x$. The $N \times N$ identity matrix is denoted by $I_N$. We let $\mathbb{1}_N$ be a vector of size $N$ with all entries equal to one. The Kronecker product is denoted by $\otimes$. We let $\mathrm{col}\{x_n\}_{n=1}^N$ denote a column vector (matrix) that stacks the vector (matrices) $x_n$ of appropriate dimensions on top of each other. The subdifferential $\partial f(x)$ of a function $f(.) : \mathbb{R}^M \to \mathbb{R}$ at some $x \in \mathbb{R}^M$ is the set of all subgradients $\partial f(x) = \{g \mid g^\mathsf{T}(y - x) \leq f(y) - f(x), \forall\, y \in \mathbb{R}^M\}$. The proximal operator with parameter $\mu > 0$ of a function $f(x) : \mathbb{R}^M \to \mathbb{R}$ is

$$\mathbf{prox}_{\mu f}(x) = \arg\min_z\ f(z) + \frac{1}{2\mu}\|z - x\|^2 \tag{5}$$

## 2 Proximal Decentralized Algorithm

In this section, we derive the algorithm and list its decentralized implementation.

### 2.1 Algorithm Derivation

We start by introducing the network weights that are used to implement the algorithm in a decentralized manner. Thus, we let $a_{sk}$ denote the weight used by agent $k$ to scale information arriving from agent $s$ with $a_{sk} = 0$ if $s$ is not a direct neighbor of agent $k$, i.e., there is no edge connecting them. Let $A = [a_{sk}] \in \mathbb{R}^{K \times K}$ denote the weight matrix associated with the network. Then, we assume $A$ to be symmetric and doubly stochastic, i.e., $A\mathbb{1}_K = \mathbb{1}_K$ and $\mathbb{1}_K^\mathsf{T} A = \mathbb{1}_K^\mathsf{T}$. We also assume that $A$ is primitive, i.e., there exists an integer $p$ such that all entries of $A^p$ are positive. Note that as long as the network is connected, there exist many ways to generate such weight matrices in a decentralized fashion – [14, 47, 48]. Under these conditions, it holds from the Perron-Frobenius theorem [49] that $A$ has a single eigenvalue at one with all other eigenvalues being strictly less than one. Therefore, $(I_K - A)x = 0$ if, and only if, $x = c\mathbb{1}_K$ for any $c \in \mathbb{R}$. If we let $w_k \in \mathbb{R}^M$ denote a local copy of the global variable $w$ available at agent $k$ and introduce the network quantities:

$$w \triangleq \mathrm{col}\{w_1, \cdots, w_K\} \in \mathbb{R}^{KM}, \quad \mathcal{B} \triangleq \frac{1}{2}(I_{KM} - A \otimes I_M) \tag{6}$$

then, it holds that $\mathcal{B}w = 0$ if, and only if, $w_k = w_s$ for all $k, s$. Note that since $A$ is symmetric with eigenvalues between $(-1, 1]$, the matrix $\mathcal{B}$ is positive semi-definite with eigenvalues in $[0, 1)$. Problem (2) is equivalent to the following constrained problem:

$$\min_{w \in \mathbb{R}^{KM}} \quad \mathcal{J}(w) + \mathcal{R}(w), \quad \text{s.t. } \mathcal{B}^{\frac{1}{2}} w = 0 \tag{7}$$

where $\mathcal{J}(w) \triangleq \sum_{k=1}^K J_k(w_k)$, $\mathcal{R}(w) \triangleq \sum_{k=1}^K R(w_k)$ and $\mathcal{B}^{\frac{1}{2}}$ is the square root of the positive semi-definite matrix $\mathcal{B}$. To solve problem (7), we introduce first the following equivalent saddle-point problem:

$$\min_w \max_y \quad \mathcal{L}_\mu(w, y) \triangleq \mathcal{J}(w) + \mathcal{R}(w) + y^\mathsf{T} \mathcal{B}^{\frac{1}{2}} w + \frac{1}{2\mu}\|\mathcal{B}^{\frac{1}{2}} w\|^2 \tag{8}$$

where $y \in \mathbb{R}^{MK}$ is the dual variable and $\mu > 0$ is the coefficient for the augmented Lagrangian. By introducing $\mathcal{J}_\mu(w) = \mathcal{J}(w) + 1/2\mu\|\mathcal{B}^{\frac{1}{2}} w\|^2$, it holds that

$$\mathcal{L}_\mu(w, y) = \mathcal{J}_\mu(w) + \mathcal{R}(w) + y^\mathsf{T} \mathcal{B}^{\frac{1}{2}} w. \tag{9}$$

To solve the saddle point problem in (8), we propose the following recursion. For $i \geq 0$:

$$\begin{cases} z_i = w_{i-1} - \mu \nabla \mathcal{J}_\mu(w_{i-1}) - \mathcal{B}^{\frac{1}{2}} y_{i-1} & \text{(10a)} \\[2mm] y_i = y_{i-1} + \alpha \mathcal{B}^{\frac{1}{2}} z_i & \text{(10b)} \\[2mm] w_i = \mathbf{prox}_{\mu \mathcal{R}}(z_i) & \text{(10c)} \end{cases}$$

where $\alpha > 0$ is the dual step-size (a tunable parameter). We will next show that with the initialization $y_0 = 0$, we can implement this algorithm in a decentralized manner.

**Remark 1** (CONVENTIONAL UPDATE). When $R(w) = 0$ and $\alpha = 1$, recursions (10a)–(10c) recover the primal-dual form of the EXTRA algorithm from [15]. However, when $R(w) \neq 0$, recursions (10a)–(10c) differ from PG-EXTRA [23] in the dual update (10b). Different from conventional dual updates that use $w_i$ (e.g., see [50] for the primal-dual form of PG-EXTRA), we use $z_i$ instead of $w_i$. This subtle difference changes the complexity of the algorithm and allows us to close the linear convergence gap between centralized and decentralized algorithms for problems of the form (2). □

## 2.2 The Decentralized Implementation

From the definition of $\mathcal{J}_\mu(w)$, we have $\nabla \mathcal{J}_\mu(w) = \nabla \mathcal{J}(w) + 1/\mu \, \mathcal{B}w$. Substituting $\nabla \mathcal{J}_\mu(w)$ into (10a), we have

$$z_i = (I_{KM} - \mathcal{B})w_{i-1} - \mu \nabla \mathcal{J}(w_{i-1}) - \mathcal{B}^{\frac{1}{2}} y_{i-1}. \tag{11}$$

With the above relation, we have for $i \geq 1$

$$z_i - z_{i-1} = (I - \mathcal{B})(w_{i-1} - w_{i-2}) - \mu\big(\nabla \mathcal{J}(w_{i-1}) - \nabla \mathcal{J}(w_{i-2})\big) - \mathcal{B}^{\frac{1}{2}}(y_{i-1} - y_{i-2}) \tag{12}$$

From (10b) we have $y_{i-1} - y_{i-2} = \alpha \mathcal{B}^{\frac{1}{2}} z_{i-1}$. Substituting this relation into (12), we reach

$$z_i = (I - \alpha\mathcal{B})z_{i-1} + (I - \mathcal{B})(w_{i-1} - w_{i-2}) - \mu\big(\nabla \mathcal{J}(w_{i-1}) - \nabla \mathcal{J}(w_{i-2})\big) \tag{13}$$

for $i \geq 1$. For initialization, we can repeat a similar argument to show that the proximal primal-dual method (10a)–(10c) with $y_0 = 0$ is equivalent to the following algorithm. Let $z_0 = w_{-1} = 0$, set $\nabla \mathcal{J}(w_{-1}) \leftarrow 0$, and $w_0$ to any arbitrary value. Repeat for $i = 1, \cdots$

$$z_i = (I - \alpha\mathcal{B})z_{i-1} + (I - \mathcal{B})(w_{i-1} - w_{i-2}) - \mu\big(\nabla \mathcal{J}(w_{i-1}) - \nabla \mathcal{J}(w_{i-2})\big) \tag{14a}$$

$$w_i = \mathbf{prox}_{\mu\mathcal{R}}(z_i) \tag{14b}$$

Since $\mathcal{B}$ has network structure, recursion (14) can be implemented in a decentralized way. This algorithm only requires each agent to share one vector at each iteration; a per agent implementation of resulting proximal primal-dual diffusion (P2D2) algorithm is listed in (15).

---

**Algorithm** (Proximal Primal-Dual Diffusion – P2D2)

**Setting**: Let $B = 0.5(I - A) = [b_{sk}]$ and choose step-sizes $\mu$ and $\alpha$. Set all initial variables to zero and repeat for $i = 1, 2, \cdots$

$$\phi_{k,i} = \sum_{s \in \mathcal{N}_k} b_{sk}(\alpha z_{s,i-1} + w_{s,i-1} - w_{s,i-2}) \qquad \textbf{(Communication Step)} \tag{15a}$$

$$\psi_{k,i} = w_{k,i-1} - \mu \nabla J_k(w_{k,i-1}) \tag{15b}$$

$$z_{k,i} = z_{k,i-1} + \psi_{k,i} - \psi_{k,i-1} - \phi_{k,i} \tag{15c}$$

$$w_{k,i} = \mathbf{prox}_{\mu R}(z_{k,i}) \tag{15d}$$

---

# 3 Main Results

In this section, we establish the linear convergence of the proximal primal-dual diffusion (P2D2) algorithm (10a)–(10c), which is equivalent to (15). To this end, we establish auxiliary lemmas leading to the main convergence result.

## 3.1 Optimality condition

We start by showing the existence and properties of a fixed point for recursions (10a)–(10c).

**Lemma 1** (FIXED POINT OPTIMILATY). *Under Assumption 1, a fixed point $(w^\star, y^\star, z^\star)$ exists for recursions* (10a)–(10c), *i.e., it holds that*

$$\begin{cases} z^\star = w^\star - \mu \nabla \mathcal{J}_\mu(w^\star) - \mathcal{B}^{\frac{1}{2}} y^\star & (16a) \\ 0 = \mathcal{B}^{\frac{1}{2}} z^\star & (16b) \\ w^\star = \mathbf{prox}_{\mu\mathcal{R}}(z^\star) & (16c) \end{cases}$$

*Moreover, $w^\star$ and $z^\star$ are unique and each block element of $w^\star = \text{col}\{w_1^\star, \cdots, w_K^\star\}$ coincides with the unique solution $w^\star$ to problem (2), i.e., $w_k^\star = w^\star$ for all $k$.*

*Proof.* The existence of a fixed point is shown in Section A in the supplementary material. We now establish the optimality of $w^\star$. Since $z^\star$ satisfies (16b), it holds that the block elements of $z^\star$ are equal to each other, i.e. $z_1^\star = \cdots = z_K^\star$, and we denote each block element by $z^\star$. Therefore, from (16c) and the definition of the proximal operator it holds that

$$w_k^\star = \arg\min_{w_k} \{R(w_k) + \|w_k - z^\star\|^2/2\mu\} \tag{17}$$

where we used $z_k^\star = z^\star$ for each $k$. Thus, we must have $w_1^\star = \cdots = w_K^\star$. We denote $w_k^\star = w^\star$ for any $k$. It is easy to verify that (17) implies

$$0 \in \partial R(w^\star) + (w^\star - z^\star)/\mu. \tag{18}$$

Multiplying $(\mathbb{1}_K \otimes I_M)^{\mathsf{T}}$ from the left to both sides of equation (16a), we get

$$Kz^\star = Kw^\star - \mu \sum_{k=1}^{K} \nabla J_k(w^\star). \tag{19}$$

Combining (18) and (19), we get $0 \in \frac{1}{K}\sum_{k=1}^{K} \nabla J_k(w^\star) + \partial R(w^\star)$, which implies that $w^\star$ is the unique solution to problem (2). Due to the uniqueness of $w^\star$, we see from (19) that $z^\star$ is unique. Consequently, $w^\star = \mathbb{1}_K \otimes w^\star$ and $z^\star = \mathbb{1}_K \otimes z^\star$ must be unique. □

**Remark 2** (PARTICULAR FIXED POINT). From Lemma 1, we see that although $w^\star$ and $z^\star$ are unique, there can be multiple fixed points. This is because from (16a), $y^\star$ is not unique due the rank deficiency of $\mathcal{B}^{\frac{1}{2}}$. However, by following similar arguments to the ones from [18], it can be verified that there exists a particular fixed point $(w^\star, y_b^\star, z^\star)$ satisfying (16a)–(16c) where $y_b^\star$ is a unique vector that belongs to the range space of $\mathcal{B}^{\frac{1}{2}}$. In the following we will show that the iterates $(w_i, y_i, z_i)$ converge linearly to this particular fixed point $(w^\star, y_b^\star, z^\star)$. □

### 3.2 Linear Convergence

To establish the linear convergence of the proximal primal-dual diffusion (P2D2) (10a)–(10c) we introduce the error quantities:

$$\widetilde{w}_i \triangleq w_i - w^\star, \quad \widetilde{y}_i \triangleq y_i - y_b^\star, \quad \widetilde{z}_i = z_i - z^\star \tag{20}$$

By subtracting (16a)–(16c) from (10a)–(10c) with $y^\star = y_b^\star$, we reach the following error recursions

$$\begin{cases} \widetilde{z}_i = \widetilde{w}_{i-1} - \mu\big(\nabla\mathcal{J}_\mu(w_{i-1}) - \nabla\mathcal{J}_\mu(w^\star)\big) - \mathcal{B}^{\frac{1}{2}}\widetilde{y}_{i-1} & (21a) \\ \widetilde{y}_i = \widetilde{y}_{i-1} + \alpha\mathcal{B}^{\frac{1}{2}}\widetilde{z}_i & (21b) \\ \widetilde{w}_i = \mathbf{prox}_{\mu\mathcal{R}}(z_i) - \mathbf{prox}_{\mu\mathcal{R}}(z^\star) & (21c) \end{cases}$$

We let $\sigma_{\max}$ and $\underline{\sigma}$ denote the maximum singular value and minimum non-zero singular value of the matrix $\mathcal{B}$. Notice that from (6), $\mathcal{B}$ is symmetric and, thus, its singular values are equal to its eigenvalues and are in $[0, 1)$ (i.e., $\sigma_{\min} = 0 < \underline{\sigma} \le \sigma_{\max} < 1$). The following result follows from [15, Proposition 3.6].

**Lemma 2** (AUGMENTED COST). *Under Assumption 1, the penalized augmented cost $\mathcal{J}(w) + \frac{\rho}{2}\|w\|_\mathcal{B}^2$ with any $\rho > 0$ is restricted strongly-convex with respect to $w^\star$:*

$$(w - w^\star)^{\mathsf{T}}\big(\nabla\mathcal{J}(w) - \nabla\mathcal{J}(w^\star)\big) + \rho\|w - w^\star\|_\mathcal{B}^2 \ge \nu_\rho\|w - w^\star\|^2 \tag{22}$$

*where*

$$\nu_\rho = \min\left\{\nu - 2\delta c, \frac{\rho\underline{\sigma}(\mathcal{B})c^2}{4(c^2+1)}\right\} > 0, \quad \text{for any } c \in \left(0, \frac{\nu}{2\delta}\right) \tag{23}$$

*for any $w$ with $w^\star = \mathbb{1} \otimes w^\star$ and where $w^\star$ denotes the minimizer of (2).*

Using the previous result, the following lemma establishes a useful inequality for later use.

**Lemma 3** (DESCENT INEQUALITY). *Under Assumption 1 and step-size conditions $\mu < \frac{(1-\sigma_{\max})}{\delta}$ and $\alpha \leq 1$, it holds that*

$$\left\|\widetilde{w}_{i-1} - \mu\left(\nabla\mathcal{J}_\mu(w_{i-1}) - \nabla\mathcal{J}_\mu(w^\star)\right)\right\|^2 \leq \gamma_1\|\widetilde{w}_{i-1}\|_{\mathcal{Q}}^2 \tag{24}$$

*where $\mathcal{Q} = I - \alpha\mathcal{B}$ and $\gamma_1 = 1 - \mu\nu_\rho(2 - \sigma_{\max} - \mu\delta) < 1$ for some $\rho > 0$ with $\nu_\rho$ given in (23).*

*Proof.* Since $\nabla\mathcal{J}_\mu(w) = \nabla\mathcal{J}(w) + \frac{1}{\mu}\mathcal{B}w$, it holds that

$$\left\|\widetilde{w}_{i-1} - \mu\left(\nabla\mathcal{J}_\mu(w_{i-1}) - \nabla\mathcal{J}_\mu(w^\star)\right)\right\|^2$$
$$= \|\widetilde{w}_{i-1}\|^2 - 2\mu\widetilde{w}_{i-1}^\mathsf{T}\left(\nabla\mathcal{J}(w_{i-1}) - \nabla\mathcal{J}(w^\star)\right) - 2\|\widetilde{w}_{i-1}\|_{\mathcal{B}}^2 + \mu^2\|\nabla\mathcal{J}_\mu(w_{i-1}) - \nabla\mathcal{J}_\mu(w^\star)\|^2 \tag{25}$$

Note that $\nabla\mathcal{J}(w) + \frac{1}{\mu}\mathcal{B}w$ is $\delta_\mu = \delta + \frac{1}{\mu}\sigma_{\max}$-Lipschitz, thus it holds that [51, Theorem 2.1.5]:

$$\|\nabla\mathcal{J}_\mu(w_{i-1}) - \nabla\mathcal{J}_\mu(w^\star)\|^2 = \|\nabla\mathcal{J}(w_{i-1}) - \nabla\mathcal{J}(w^\star) + \frac{1}{\mu}\mathcal{B}\widetilde{w}_{i-1}\|^2$$

$$\leq \delta_\mu\widetilde{w}_{i-1}^\mathsf{T}\left(\nabla\mathcal{J}(w_{i-1}) - \nabla\mathcal{J}(w^\star) + \frac{1}{\mu}\mathcal{B}\widetilde{w}_{i-1}\right) \tag{26}$$

Substituting the previous inequality into (25) we get

$$\left\|\widetilde{w}_{i-1} - \mu\left(\nabla\mathcal{J}_\mu(w_{i-1}) - \nabla\mathcal{J}_\mu(w^\star)\right)\right\|^2$$
$$\leq \|\widetilde{w}_{i-1}\|^2 - \mu(2 - \mu\delta_\mu)\widetilde{w}_{i-1}^\mathsf{T}\left(\nabla\mathcal{J}(w_{i-1}) - \nabla\mathcal{J}(w^\star)\right) - (2 - \mu\delta_\mu)\|\widetilde{w}_{i-1}\|_{\mathcal{B}}^2$$
$$\leq \left(1 - \mu\nu_\rho(2 - \mu\delta_\mu)\right)\|\widetilde{w}_{i-1}\|^2 - (2 - \mu\delta_\mu)(1 - \rho\mu)\|\widetilde{w}_{i-1}\|_{\mathcal{B}}^2 \tag{27}$$

where the last inequality follows from (22) and $2 - \mu\delta_\mu = 2 - \sigma_{\max} - \mu\delta > 0$ for $\mu < \frac{2-\sigma_{\max}}{\delta}$. Letting $\gamma_1 = 1 - \mu\nu_\rho(2 - \mu\delta_\mu)$ and adding and subtracting $\alpha\gamma_1\|\widetilde{w}_{i-1}\|_{\mathcal{B}}$ to the right hand side of the previous inequality gives:

$$\left\|\widetilde{w}_{i-1} - \mu\left(\nabla\mathcal{J}_\mu(w_{i-1}) - \nabla\mathcal{J}_\mu(w^\star)\right)\right\|^2 \leq \gamma_1\|\widetilde{w}_{i-1}\|_{\mathcal{Q}}^2 - \left((2 - \mu\delta_\mu)(1 - \rho\mu) - \alpha\gamma_1\right)\|\widetilde{w}_{i-1}\|_{\mathcal{B}}^2 \tag{28}$$

where $\mathcal{Q} = I - \alpha\mathcal{B}$. If we can ensure that

$$-\left((2 - \mu\delta_\mu)(1 - \mu\rho) - \alpha\gamma_1\right)\|\widetilde{w}_{i-1}\|_{\mathcal{B}}^2 \leq 0 \tag{29}$$

then inequality (28) can be upper bounded by (24). To ensure inquality (29), it is sufficient to find $\mu$ and $\rho$ such that

$$(2 - \mu\delta_\mu)(1 - \mu\rho) - \gamma_1\alpha = (2 - \sigma_{\max} - \mu\delta)(1 - \mu\rho) - \gamma_1\alpha \geq 0 \tag{30}$$

By noting that $\gamma_1 = 1 - \mu\nu_\rho(2 - \sigma_{\max} - \mu\delta) < 1$ for $\mu < \frac{1-\sigma_{\max}}{\delta}$ and using $\alpha \leq 1$, the above inequality is guaranteed to hold if

$$0 < \rho \leq \frac{1 - \sigma_{\max} - \mu\delta}{\mu(2 - \sigma_{\max} - \mu\delta)} \tag{31}$$

□

The previous lemma will be used to establish the following primal-dual error bound.

**Lemma 4** (ERROR BOUND). *Under Assumption 1, if $y_0 = 0$ and the step-sizes satisfy $\mu < \frac{(1-\sigma_{\max})}{\delta}$ and $\alpha \leq 1$, it holds that*

$$\|\widetilde{z}_i\|_{\mathcal{Q}}^2 + \|\widetilde{y}_i\|_{\frac{1}{\alpha}I}^2 \leq \gamma_1\|\widetilde{w}_{i-1}\|_{\mathcal{Q}}^2 + \gamma_2\|\widetilde{y}_{i-1}\|_{\frac{1}{\alpha}I}^2 \tag{32}$$

*where $\mathcal{Q} = I - \alpha\mathcal{B} > 0$, $\gamma_1 = 1 - \mu\nu_\rho(2 - \sigma_{\max} - \mu\delta) < 1$, and $\gamma_2 = 1 - \alpha\underline{\sigma} < 1$ for some $\rho > 0$ with $\nu_\rho$ given in (23).*

*Proof.* Squaring both sides of (21a) and (21b) we get

$$\|\widetilde{z}_i\|^2 = \|\widetilde{w}_{i-1} - \mu\big(\nabla\mathcal{J}_\mu(w_{i-1}) - \nabla\mathcal{J}_\mu(w^\star)\big)\|^2 + \|\mathcal{B}^{\frac{1}{2}}\widetilde{y}_{i-1}\|^2$$
$$- 2\widetilde{y}_{i-1}^\mathsf{T}\mathcal{B}^{\frac{1}{2}}\left(\widetilde{w}_{i-1} - \mu\big(\nabla\mathcal{J}_\mu(w_{i-1}) - \nabla\mathcal{J}_\mu(w^\star)\big)\right) \tag{33}$$

and

$$\|\widetilde{y}_i\|^2 = \|\widetilde{y}_{i-1} + \alpha\mathcal{B}^{\frac{1}{2}}\widetilde{z}_i\|^2 = \|\widetilde{y}_{i-1}\|^2 + \alpha^2\|\mathcal{B}^{\frac{1}{2}}\widetilde{z}_i\|^2 + 2\alpha\widetilde{y}_{i-1}^\mathsf{T}\mathcal{B}^{\frac{1}{2}}\widetilde{z}_i$$
$$\overset{(21a)}{=} \|\widetilde{y}_{i-1}\|^2 + \alpha^2\|\widetilde{z}_i\|_\mathcal{B}^2 - 2\alpha\|\mathcal{B}^{\frac{1}{2}}\widetilde{y}_{i-1}\|^2$$
$$+ 2\alpha\widetilde{y}_{i-1}^\mathsf{T}\mathcal{B}^{\frac{1}{2}}\left(\widetilde{w}_{i-1} - \mu\big(\nabla\mathcal{J}_\mu(w_{i-1}) - \nabla_w\mathcal{J}_\mu(w^\star)\big)\right) \tag{34}$$

Multiplying equation (34) by $\frac{1}{\alpha}$ and adding to (33), we get

$$\|\widetilde{z}_i\|_\mathcal{Q}^2 + \|\widetilde{y}_i\|_{\frac{1}{\alpha}I}^2 = \|\widetilde{w}_{i-1} - \mu\big(\nabla\mathcal{J}_\mu(w_{i-1}) - \nabla\mathcal{J}_\mu(w^\star)\big)\|^2 + \|\widetilde{y}_{i-1}\|_{\frac{1}{\alpha}I}^2 - \alpha\|\mathcal{B}^{\frac{1}{2}}\widetilde{y}_{i-1}\|_{\frac{1}{\alpha}I}^2 \tag{35}$$

where $\mathcal{Q} = I - \alpha\mathcal{B}$. Since both $y_i$ and $y_b^\star$ lie in the range space[3] of $\mathcal{B}^{\frac{1}{2}}$, it holds that $\|\mathcal{B}^{\frac{1}{2}}\widetilde{y}_{i-1}\|^2 \geq \underline{\sigma}\|\widetilde{y}_{i-1}\|^2$ – see [18]. Thus, we can bound (35) by

$$\|\widetilde{z}_i\|_\mathcal{Q}^2 + \|\widetilde{y}_i\|_{\frac{1}{\alpha}I}^2 \leq \|\widetilde{w}_{i-1} - \mu\big(\nabla\mathcal{J}_\mu(w_{i-1}) - \nabla\mathcal{J}_\mu(w^\star)\big)\|^2 + (1 - \alpha\underline{\sigma})\|\widetilde{y}_{i-1}\|_{\frac{1}{\alpha}I}^2 \tag{36}$$

Under the conditions given in Lemma 3, we can substitute inequality (24) in the above relation and get (32). Note that that $\gamma_1 = 1 - \mu\nu_\rho(2 - \sigma_{\max} - \mu\delta) < 1$ if $\mu < (1 - \sigma_{\max})/\delta$. Moreover, $\mathcal{Q} = I - \alpha\mathcal{B} > 0$ and $\gamma_2 = 1 - \alpha\underline{\sigma} < 1$ for $\alpha \leq 1$ since $\sigma_{\max} < 1$. □

The next Theorem establishes the linear convergence of our proposed algorithm.

**Theorem 1** (LINEAR CONVERGENCE). *Under Assumption 1, $y_0 = 0$, and if step-sizes satisfy*

$$\mu < \frac{(1 - \sigma_{\max})}{\delta}, \quad \alpha \leq \min\left\{1, \mu\nu_\rho(2 - \sigma_{\max} - \mu\delta)\right\}. \tag{37}$$

*It holds that $\|\widetilde{w}_i\|^2 \leq C\gamma^i$ where $C > 0$ and*

$$\gamma \overset{\Delta}{=} \max\left\{1 - \mu\nu_\rho(2 - \sigma_{\max} - \mu\delta)/(1 - \alpha\sigma_{\max}), 1 - \alpha\underline{\sigma}\right\} < 1 \tag{38}$$

*for some $\rho > 0$ with $\nu_\rho$ given in (23).*

*Proof.* Assume $\alpha \leq 1$ and note that $\mathcal{Q} = I - \alpha\mathcal{B}$. Thus, it holds that $\sigma_{\min}(\mathcal{Q}) = 1 - \alpha\sigma_{\max}$ and $\sigma_{\max}(\mathcal{Q}) = 1$. This implies that $(1 - \alpha\sigma_{\max})\|x\|^2 \leq \|x\|_\mathcal{Q}^2 \leq \|x\|^2$ for any $x \in \mathbb{R}^{KM}$. Therefore, it holds from Lemma 4 that

$$(1 - \alpha\sigma_{\max})\|\widetilde{z}_i\|^2 + \|\widetilde{y}_i\|_{\frac{1}{\alpha}I}^2 \leq \gamma_1\|\widetilde{w}_{i-1}\|^2 + \gamma_2\|\widetilde{y}_{i-1}\|_{\frac{1}{\alpha}I}^2 \tag{39}$$

when $\mu < \frac{(1-\sigma_{\max})}{\delta}$. Dividing by $\beta \overset{\Delta}{=} 1 - \alpha\sigma_{\max}$ both sides of the above inequality, we have

$$\|\widetilde{z}_i\|^2 + \|\widetilde{y}_i\|_{\frac{1}{\alpha\beta}I}^2 \leq \frac{\gamma_1}{\beta}\|\widetilde{w}_{i-1}\|^2 + \gamma_2\|\widetilde{y}_{i-1}\|_{\frac{1}{\alpha\beta}I}^2. \tag{40}$$

Clearly, $\beta \in (0, 1)$ when $\alpha \leq 1$. Now we evaluate $\gamma_1/\beta$. It is easy to verify that

$$\frac{\gamma_1}{\beta} = (1 - \mu\nu_\rho(2 - \sigma_{\max} - \mu\delta))/(1 - \alpha\sigma_{\max}) < 1 \tag{41}$$

when $\alpha \leq \mu\nu_\rho(2 - \sigma_{\max} - \mu\delta) < \frac{\mu\nu_\rho(2-\sigma_{\max}-\mu\delta)}{\sigma_{\max}}$. Next, from the non-expansiveness property of the proximal operator we have:

$$\|\widetilde{w}_i\|^2 = \|\mathbf{prox}_{\mu\mathcal{R}}(z_i) - \mathbf{prox}_{\mu\mathcal{R}}(z^\star)\|^2 \leq \|\widetilde{z}_i\|^2. \tag{42}$$

By substituting (42) into (40) and letting $\gamma \overset{\Delta}{=} \max\{\gamma_1/\beta, \gamma_2\}$, we reach

$$\|\widetilde{w}_i\|^2 + \|\widetilde{y}_i\|_{\frac{1}{\alpha\beta}I}^2 \leq \gamma\left(\|\widetilde{w}_{i-1}\|^2 + \|\widetilde{y}_{i-1}\|_{\frac{1}{\alpha\beta}I}^2\right) \tag{43}$$

when step-sizes $\mu$ and $\alpha$ satisfy condition (37). We iterate the above inequality and get

$$\|\widetilde{w}_i\|^2 \leq \|\widetilde{w}_i\|^2 + \|\widetilde{y}_i\|_{\frac{1}{\alpha\beta}I}^2 \leq \gamma^i(\|\widetilde{w}_0\|^2 + \|\widetilde{y}_0\|_{\frac{1}{\alpha\beta}I}^2), \tag{44}$$

which concludes the proof. □

Next we show that when $R(w) = 0$, we can have a better upper bound for the dual step-size, which covers the EXTRA algorithm [15].

**Theorem 2** (LINEAR CONVERGENCE WHEN $R(w) = 0$)**.** *Under Assumption 1, if $R(w) = 0$, $y_0 = 0$, and the step-sizes satisfy $\mu < \frac{(1 - \sigma_{\max})}{\delta}$ and $\alpha \leq 1$, it holds that $\|\widetilde{w}_i\|_{\mathcal{Q}}^2 \leq C\gamma^i$ where $C > 0$, $\mathcal{Q} = I - \alpha\mathcal{B} > 0$, and*

$$\gamma = \max\left\{1 - \mu\nu_\rho(2 - \sigma_{\max} - \mu\delta), 1 - \alpha\underline{\sigma}\right\} < 1$$

*for some $\rho > 0$ with $\nu_\rho$ given in (23).*

*Proof.* From lemma 4, we know when $\mu < \frac{(1 - \sigma_{\max})}{\delta}$ and $\alpha \leq 1$ that inequality (32) holds. Since $R(w) = 0$, we know $w_i = z_i$ from recursion (10c) and hence $\widetilde{w}_i = \widetilde{z}_i$. By letting $\gamma = \max\{\gamma_1, \gamma_2\}$, inequality (32) becomes $\|\widetilde{w}_i\|_{\mathcal{Q}}^2 + \|\widetilde{y}_i\|_{\frac{1}{\alpha}I}^2 \leq \gamma(\|\widetilde{w}_{i-1}\|_{\mathcal{Q}}^2 + \|\widetilde{y}_{i-1}\|_{\frac{1}{\alpha}I}^2)$. Since $\mathcal{Q} = I - \alpha\mathcal{B}$ is positive definite when $\alpha \leq 1$, we reach the linear convergence of $\widetilde{w}_i$. $\square$

In the above Theorem, we see that the convergence rate bound is upper bounded by two terms, one term is from the cost function and the other is from the network. This bound shows how the network affects the convergence rate of the algorithm. For example, in Theorem 2, assume that $\alpha = 1$ and the network term dominates the convergence rate so that $\gamma = 1 - \alpha\underline{\sigma} = 1 - \underline{\sigma}$. Recall that $\underline{\sigma} = \underline{\sigma}(\mathcal{B})$ is the smallest non-zero singular value (or eigenvalue) of the matrix $0.5(I - A)$. Thus, the effect of the network on the convergence rate is evident through the term $1 - \underline{\sigma}$, which becomes close to one as the network becomes more sparse. Note when $\alpha = 1$, the algorithm recovers EXTRA as highlighted in Remark 1. In this case, our step-size condition is on the order of $O((1 - \sigma_{\max})/\delta)$. Note that the in the original EXTRA proof in [15, Theorem 3.7], the step-size bound is on the order of $O(\nu_\rho(1 - \sigma_{\max})/\delta^2))$, which scales badly for ill-conditioned problems, i.e., if $\delta$ is much larger than $\nu_\rho$. We remark that simulations of the proposed algorithm are provided in Section B in the supplementary material.

**Acknowledgments**

This work was supported in part by NSF grant CCF-1524250. We would like to thank the anonymous reviewers for their insightful comments.

## Footnotes

[1] The function $f(.)$ is proper if $-\infty < f(x)$ for all $x$ in its domain and $f(x) < \infty$ for at least one $x$.

[2] A sequence $\{x_i\}_{i=0}^{\infty}$ has asymptotic linear convergence to $x^\star$ if there exists a sufficiently large $i_o$ such that $\|x_i - x^\star\| \le \gamma^i C$ for some $C > 0$ and all $i \ge i_o$.

[3] Since $y_0 = 0$ and $y_i = y_{i-1} + \alpha\mathcal{B}^{\frac{1}{2}}z_i$, we know $y_i \in \mathrm{range}(\mathcal{B}^{\frac{1}{2}})$ for any $i$.

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
