[Supplementary Material]

### SUPPLEMENTARY MATERIAL for "A Linearly Convergent Proximal Gradient Algorithm for Decentralized Optimization"

## A  Existence of a Fixed Point Proof for Lemma 1

To establish existence we will construct a point $(\mathcal{w}^\star, \mathcal{y}^\star, \mathcal{z}^\star)$ that satisfies equations (16a)–(16c). From assumption (1), there exists a unique solution $w^\star$ for problem (2). From the optimality condition, there must exist a subgradient $r^\star \in \partial R(w^\star)$ such that

$$\frac{1}{K}\sum_{k=1}^{K} \nabla J_k(w^\star) + r^\star = 0 \tag{45}$$

We see from the above equation that $r^\star$ is unique due to the uniqueness of $w^\star$. Now define $z^\star \triangleq \mu r^\star + w^\star$. It holds that $(z^\star - w^\star) = \mu r^\star$, i.e., $(z^\star - w^\star) \in \mu \partial R(w^\star)$. This implies that

$$w^\star = \arg\min_{w} \left\{ R(w) + \frac{1}{2\mu}\|w - z^\star\|^2 \right\}. \tag{46}$$

We next define $\mathcal{w}^\star = \mathbb{1}_K \otimes w^\star$ and $\mathcal{z}^\star = \mathbb{1}_K \otimes z^\star$. Relation (46) implies that equation (16c) holds. Also, since $\mathcal{z}^\star = \mathbb{1}_K \otimes z^\star$, it belongs to the null space of $\mathcal{B}^{\frac{1}{2}}$ so that $\mathcal{B}^{\frac{1}{2}} \mathcal{z}^\star = 0$. It remains to construct $\mathcal{y}^\star$ that satisfies equation (16a). Note that $\nabla \mathcal{J}_\mu(\mathcal{w}^\star) = \nabla \mathcal{J}(\mathcal{w}^\star) + \frac{1}{\mu}\mathcal{B}\mathcal{w}^\star = \nabla \mathcal{J}(\mathcal{w}^\star)$ due to the fact that $\mathcal{w}^\star$ lies in the null space of $\mathcal{B}$, and therefore

$$(\mathbb{1}_N \otimes I_M)^\mathsf{T}\big(\mathcal{w}^\star - \mathcal{z}^\star - \mu\nabla\mathcal{J}_\mu(\mathcal{w}^\star)\big) = -\mu K r^\star - \mu \sum_{k=1}^{K}\nabla J_k(w^\star) \overset{(45)}{=} 0, \tag{47}$$

where the last equality holds because of (45). Equation (47) implies that

$$\big(\mathcal{w}^\star - \mathcal{z}^\star - \mu\nabla\mathcal{J}_\mu(\mathcal{w}^\star)\big) \in \mathrm{Null}(\mathbb{1}_N \otimes I_M) = \mathrm{Null}(\mathcal{B}^{\frac{1}{2}})^\perp = \mathrm{Range}(\mathcal{B}^{\frac{1}{2}}) \tag{48}$$

where $\perp$ denotes the orthogonal complement of a set. As a result, there must exist a vector $\mathcal{y}^\star$ that satisfies equation (16a).

## B  Numerical Simulations

In this section we verify our results with numerical simulations. We consider the decentralized sparse logistic regression problem for three real datasets[4]: Covtype.binary, MNIST, and CIFAR10. The last two datasets have been transformed into binary classification problems by considering digital two and four ('2' and '4') classes for MNIST, and cat and dog classes for CIFAR-10. In Covtype.binary we used 50,000 samples as training data and each data has dimension 54. We used 10,000 samples as training data from MNIST (with labels '2' and '4') and each data has dimension 784. In CIFAR-10 we used 10,000 training data (with labels 'cat' and 'dog') and each data has dimension 3072. All features have been preprocessed by normalizing them to the unit vector with sklearn's normalizer[5]. For the network, we generated a randomly connected network with $K = 20$ nodes – see Fig. 1. The associated combination matrix $A$ is generated according to the Metropolis rule [14, 47]. The decentralized sparse logistic regression problem takes the form

$$\min_{w\in\mathbb{R}^M} \frac{1}{K}\sum_{k=1}^{K} J_k(w) + \rho\|w\|_1 \quad \text{where} \quad J_k(w) = \frac{1}{L}\sum_{\ell=1}^{L}\ln(1+\exp(-y_{k,\ell}x_{k,\ell}^\mathsf{T}w)) + \frac{\lambda}{2}\|w\|^2$$

where $\{x_{k,\ell}, y_{k,\ell}\}_{\ell=1}^{L}$ are local data kept by agent $k$ and $L$ is the size of the local dataset. For all simulations, we assign data samples evenly to each local agent. We set $\lambda = 10^{-4}$ and $\rho = 0.002$ for Covtype, $\lambda = 10^{-2}$ and $\rho = 0.0005$ for CIFAR-10, and $\lambda = 10^{-4}$ and $\rho = 0.002$ for MNIST. We compare the proposed P2D2 method against two well-know proximal gradient-based decentralized algorithms that can handle non-smooth regularization terms: PG-EXTRA [23] and decentralized

linearized ADMM (DL-ADMM) [22, 42]. For each algorithm, we tune the step-size to the best possible convergence rate. The step-sizes employed in each method for each dataset are listed in Table 1. Also, the proposed method employs an additional step-size $\alpha$ which is set as $1, 0.8$ and $1$ for Covtype, CIFAR-10 and MNIST, respectively. Figure 2 shows that each local variable $w_{k,i}$ converges linearly to the global solution $w^\star$ for the proposed method (14a)–(14b), which is consistent with Theorem 1. The proposed method is slightly faster than DL-ADMM and PG-EXTRA due to the additional tunable parameter $\alpha$. Note that while DL-ADMM and PG-EXTRA are observed to converge linearly, no theoretical guarantees are shown in [22, 23, 42]. The simulation code is provided in the supplementary material.

| | Covtype | CIFAR-10 | MNIST |
|---|---|---|---|
| DL-ADMM | 0.0022 | 0.075 | 0.21 |
| PG-EXTRA | 0.002 | 0.07 | 0.20 |
| P2D2 (Proposed) | 0.0024 | 0.08 | 0.24 |

**Table 1:** Step-sizes used in the simulation.

**Figure 1:** The network topology used in the simulation.

**Figure 2:** Simulation Results. The $y$-axis indicates the relative squared error $\sum_{k=1}^{K} \|w_{k,i} - w^\star\|^2 / \|w^\star\|^2$.

## Footnotes

[4]Covtype: www.csie.ntu.edu.tw, MNIST: yann.lecun.com, CIFAR10: www.cs.toronto.edu.

[5]https://scikit-learn.org