[Reviews · NeurIPS 2019]

Reviewer 1



[I read the response, no change in my opinion, mostly due to limitation of my confidence. I at least tried to challenge other reviewers in discussion...] Within the constraint of NeurIPS format, I think the authors do a very good job at building the intuition for the complex proofs. Overall, the result *look like* roughly what I would expect, it is good to see clearly the dependence of network structure in convergence bounds, and think the result valuable. However, I am concerned that this work will not be properly reviewed in this conference. (I did not check the proofs) Section 4 To be honest, I would prefer removing the whole section. The experiments do not reflect anything practical at the moment. The contribution is clearly theoretical, addresses a (in my opinion) interesting problem, but I am not aware of a single production use - but I can surely imagine some in the future enabled by progress in telecommunications technology. Other L142 - Is p and integer? make clear L152 / Eq (7) - Provide a reference? I thought some of the remarks in early Sec 2 (ref [43-45], Problem reformulation) could be relevant to work of Loizou and Richtarik: A new perspective on randomized gossip algorithms, which you are maybe unaware of.

Reviewer 2



First, I would like to commend the authors on a very nice paper. I found the paper to be exceptionally clear and well-written. Moreover, the introduction provides a clear and thorough review of the related literature. The results are clean, and I believe are of great theoretical value. Overall, I think this is a very nice paper. I only have a few comments: 1) In Remark 1 (lines 160-168), the authors state that "the DIGing algorithm can also be related to EXTRA [...] our technique covers that form of DIGing". I feel this is a misleading statement. DIGing is a fundamentally different algorithm from EXTRA in that it requires two communications through the graph in each iteration. In the DIGing paper (ref [2]), when the authors mention that DIGing is related to EXTRA, their precise statement was that if you choose the two mixing matrices of EXTRA in a particular way. i.e. W1 = 2W-I and W2 = W^2, then it BECOMES DIGing (note that choosing W2=W^2 means that you're communicating twice). Returning to the present manuscript, it is therefore meaningless to say that the new algorithm covers both EXTRA and also "that form of DIGing", since that form of DIGing is identically equivalent to EXTRA. 2) In the discussion that follows Theorem 2 (comparison to EXTRA in the case R=0), the authors claim that they provide a tighter bound on the stepsize mu than the bound in the original EXTRA paper (ref [1]). Again, I think this statement needs more qualifications. The two papers make different assumptions about the objective functions and therefore the results are not directly comparable. Specifically, the present paper assumes each J_k is smooth+strongly convex and that R is convex. Meanwhile, the result from the EXTRA paper only assumes that the global objective satisfies a restricted strong convexity property (the individual J_k need not). Since the present manuscript makes stronger assumptions than the EXTRA paper did, one would expect the bound to be tighter. 3) In the numerical simulations (Section 4), the authors state that PG-EXTRA is observed to converge linearly, but there is no theoretical guarantee that they in fact do. I am not familiar with PG-EXTRA, but since the proposed algorithm subsumes EXTRA in the R=0 case, does it also subsume PG-EXTRA in the R != 0 case? It would be worth briefly discussing this, since EXTRA and PG-EXTRA have such similar names and were developed by the same authors.

Reviewer 3



This paper modifies the primal-dual form of EXTRA and shows the linear convergence of the algorithm under certain assumptions, which partially closes the gap between decentralized algorithms and centralized algorithms with the presence of the non-smooth term in theory. It also extends the convergence results of EXTRA under a stronger assumption. The proof is concise and correct. One drawback is the strong assumption it builds on. It would be helpful if this assumption can be removed or weakened.

Reviewer 4



The authors provided a proof for the linear convergence rate of the relaxed EXTRA algorithm for synchronous decentralized consensus optimization. Modifications to existing methods are minor since the same relaxation and additional non-smooth regularization have been used on the same problem in the literature. The contribution is therefore in the proof of the linear rate under the assumption of strong convexity of each local objective function. The proof technique is standard as appeared in related works. Numerical experiments are carried out for a decentralized logistic regression problem. The contribution of this work is incremental but not significant. The proof requires strong convexity of all local objectives and synchronized communications which are very strict in practice. -- After response I read the response, but I'm still not convinced by the novelty and significance of this work. Similar linear convergence rate has been established for EXTRA and decentralized linearized ADMM in the literature, from which the proposed algorithm adopted the main structure. The additional extrapolation and non-smooth convex R (when handled by proximity operator) are only minor changes from the existing method and do not introduce much difficulty in convergence analysis from the optimization point of view. The significance to the ML community is also limited with the (restricted) strong convexity assumption and synchronous setting. I also disagree with the authors on their claim that the analysis in their work can be used to prove linear convergence rate for asynchronous setting, which is greatly complicated than the synchronous setting due to the excessive randomness in communications etc.

[Author Response · NeurIPS 2019]

[General response] We thank the reviewers for their useful comments. Before we go through our specific response to
each reviewer, we would like to make two general comments that can be useful for all reviewers. **First**, The results from
the paper by Woodworth and Srebro (NIPS 2016) [arxiv:1605.08003] imply that decentralized proximal algorithms
cannot achieve linear convergence in the presence of more than one non-smooth term in the worst case. This conclusion
illustrates the importance of the global common regularizer structure in our problem set-up. **Second**, We can *relax*
the strong-convexity assumption and extend our analysis to cover the scenario in which the aggregate cost function
$\bar{J}(w) = \frac{1}{K} \sum_{k=1}^{K} J_k(w) : \mathbb{R}^M \to \mathbb{R}$ is restricted strongly-convex (see [1] for details) while each individual cost
$J_k(w)$ is not necessarily convex. This extension requires minor modifications and can be included in the revised
manuscript. Let us briefly explain how to extend our analysis to this case. We know from the EXTRA paper [1] that
the aggregate cost $\bar{J}(w)$ is $\bar{\nu}$-restricted-strongly-convex if, and only if, the penalized augmented cost $\mathcal{J}(w) + \frac{\rho}{2}\|w\|_{\mathcal{B}}^2$
$(\mathcal{J}(w) = \frac{1}{K} \sum_{k=1}^{K} J_k(w_k) : \mathbb{R}^{MK} \to \mathbb{R})$ is $\bar{\nu}_\rho$-restricted-strongly-convex for some $\bar{\nu}_\rho > 0$ and any scalar $\rho > 0$.
This means that for any $w$, it holds [1]: $(w - w^\star)^\mathsf{T}(\nabla \mathcal{J}(w) - \nabla \mathcal{J}(w^\star)) + \rho\|w - w^\star\|_{\mathcal{B}}^2 \geq \bar{\nu}_\rho \|w - w^\star\|^2$. Since
$\mathcal{J}_\mu(w) = \mathcal{J}(w) + \frac{1}{2\mu}\|w\|_{\mathcal{B}}^2$ has $\delta_\mu = (\delta + \frac{1}{\mu}\sigma_{\max})$-Lipschitz gradients it satisfies (see inequality (2.1.8) from [46])
$\|\nabla \mathcal{J}_\mu(w) - \nabla \mathcal{J}_\mu(w^\star)\|^2 \leq \delta_\mu(\nabla \mathcal{J}_\mu(w) - \nabla \mathcal{J}_\mu(w^\star))^\mathsf{T}(w - w^\star) = \delta_\mu(\nabla \mathcal{J}(w) - \nabla \mathcal{J}(w^\star))^\mathsf{T}\widetilde{w} + \frac{1}{\mu}\delta_\mu\|\widetilde{w}\|_{\mathcal{B}}^2$.
Using the above two inequalities we reach $\|\widetilde{w}_{i-1} - \mu(\nabla \mathcal{J}_\mu(w_{i-1}) - \nabla \mathcal{J}_\mu(w^\star))\|^2 \leq (1 - \mu(2 - \mu\delta_\mu)\bar{\nu}_\rho)\|\widetilde{w}_{i-1}\|^2 -$
$(2 - \sigma_{\max} - \mu\delta - \mu\rho)\|\widetilde{w}_{i-1}\|_{\mathcal{B}}^2$. Replacing the bound in Lemma 2 by this bound and following the same proof technique
in the paper we can establish linear convergence under this more relaxed condition for some small enough constant $\rho > 0$
that depends on $\mu$ and $\sigma_{\max}$. For the smooth case, we can guarantee linear convergence for $\mu \leq O((1-\sigma_{\max})/\delta)$, which
is still tighter than EXTRA $O(\bar{\nu}_\rho(1 - \sigma_{\max})/\delta^2)$ step-size. Note that the restricted strongly-convex condition is weaker
than strong-convexity and it is met in sparse optimization settings and others – see Hui and Yin. [arXiv:1303.4645],
Hui [arXiv:1511.01635] and references therein.

[Reviewer 1] Thank you for the positive comments on the paper. We agree it is better to move the simulation section to
the supplementary material since the contributions are theoretical. In L142, $p$ is an integer, which will be highlighted in
the revision. In Eq. (7), we can cite [1] or [2]. The remarks in early section 2 are relevant to section 3 of the work of
Loizou and Richtarik, which we were not aware of. We will include this work in the reference list.

[Reviewer 2] We thank the reviewer for his/her insightful and encouraging comments. We agree with all of the reviewers'
comments and will make further clarifications in the revision: 1) DIGing can be fundamentally different from EXTRA
as highlighted by the reviewer. That said, for a static and undirected network, both DIGing and EXTRA belong to
a class of primal-descent dual-ascent methods applied to the augmented Lagrangian function. They only differ in
choosing the weight matrix and augmented Lagrangian penalty matrix, which are highlighted in the supplementary
material. To avoid misleading the readers, we can rephrase that remark by stating that our technique covers the class of
augmented Lagrangian methods without mentioning DIGing. 2) As highlighted in the general response above, we can
prove linear convergence under the same assumptions as EXTRA, and still provide tighter bounds. 3) The proposed
algorithm in this work does not cover PG-EXTRA when $R(.) \neq 0$. The difference lies in the order by which the agents
conduct their updates and the proximal mapping. We will clarify this point in the revision.

[Reviewer 4] We thank the reviewer for his/her comments and for verifying the correctness of the proof. Please see the
second point in the general response above regarding relaxing our assumption.

[Reviewer 5] We thank the reviewer for his/her comments. We respectively disagree that our contribution is incremental.
Our work closes the gap between the convergence rate of the centralized and decentralized methods for composite
optimization problems. As observed by the reviewer, with a *slight* modification (yet novel) on the algorithm structure,
we resolve a long-standing open question, which is a significant contribution in our opinion. In addition, our proof
technique delicately handles the proximal mapping and, moreover, it is shorter and more constructive than existing
proofs related to this work. For example, one can check that our proof is simpler to verify than those in [1], [2] and
[26]. Third, while non-convex and asynchronous settings are of great practical value, they are beyond the scope of this
paper. Note that the gap between the understanding of centralized and decentralized algorithms still exists even in the
synchronous and convex scenario. Any theoretical breakthrough in the synchronous and convex optimization can be
beneficial. For example, the novel work [29] closed the gap between decentralized and centralized optimal algorithms
under similar conditions to this work and left the more practical conditions for future work – see [Section 4.3, 29]. Later,
other works studied the more practical conditions – see [37] and [arXiv:1810.02660]. Another example is the work [20],
which still has impact and is useful to current decentralized optimization works. Similarly, with the idea established in
this work and [arXiv:1612.00150, arXiv:1810.02660], it is possible to establish the linear convergence for composite
problems under the asynchronous setting. Finally, to partially resolve the reviewer's suggested improvements, we can
relax the strong-convexity assumption so that only the aggregate cost, rather than each local $J_k(w)$, is required to be
restricted-strongly-convex, see point two in the general response above.

[Meta-Review · NeurIPS 2019]

Most of the reviewers like the paper a lot, however, reviewer #5 was ok with acceptance, however, he was not 100% convinced by the novelty and significance of this work. I would strongly suggest taking into account the valuable comments of the reviewers when preparing your final camera-ready submission!